# Implementation of Glucose-6-Phosphate Dehydrogenase (G6PD) testing for *Plasmodium vivax* case management, a mixed method study from Cambodia

**Sarah A. Cassidy-Seyoum**[1,2]*, **Keoratha Chheng**[3], **Phal Chanpheakdey**[3], **Agnes Meershoek**[2], **Michelle S. Hsiang**[4,5,6], **Lorenz von Seidlein**[3,7], **Rupam Tripura**[3,7], **Bipin Adhikari**[3,7], **Benedikt Ley**[1,8], **Ric N. Price**[1,3,7], **Dysoley Lek**[9,10], **Nora Engel**[2☯], **Kamala Thriemer**[1☯]

1 Global and Tropical Health Division, Menzies School of Health Research, Charles Darwin University, Darwin, Australia, 2 Department of Health Ethics and Society, Care and Public Health Research Institute (CAPHRI), Maastricht University, Maastricht, The Netherlands, 3 Faculty of Tropical Medicine, Mahidol Oxford Research Unit, Mahidol University, Bangkok, Thailand, 4 Institute for Global Health Sciences, Malaria Elimination Initiative, University of California San Francisco, San Francisco, California, United States of America, 5 Department of Epidemiology and Biostatistics, University of California San Francisco, San Francisco, California, United States of America, 6 Department of Pediatrics, Division of Pediatric Infectious Diseases, University of California San Francisco, San Francisco, California, United States of America, 7 Nuffield Department of Medicine, Center for Tropical Medicine and Global Health, University of Oxford, Oxford, United Kingdom, 8 Division of Education, Menzies School of Health Research, Charles Darwin University, Darwin, Australia, 9 National Center for Parasitology, Entomology and Malaria Control, Phnom Penh, Cambodia, 10 National Institute of Public Health, School of Public Health, Phnom Penh, Cambodia

☯ These authors contributed equally to this work.
* Sarah.Cassidy-Seyoum@menzies.edu.au

## Abstract

*Plasmodium vivax* remains a challenge for malaria elimination since it forms dormant liver stages (hypnozoites) that can reactivate after initial infection. 8-aminoquinolone drugs kill hypnozoites but can cause severe hemolysis in individuals with Glucose-6-Phosphate Dehydrogenase (G6PD) deficiency. The STANDARD G6PD test (Biosensor) is a novel point-of-care diagnostic capable of identifying G6PD deficiency prior to treatment. In 2021, Cambodia implemented the Biosensor to facilitate radical cure treatment for vivax malaria. To assess the Biosensor's implementation after its national rollout, a mixed-methods study was conducted in eight districts across three provinces in Cambodia. Interviews, focus group discussions, and observations explored stakeholders' experiences with G6PD testing and factors influencing its implementation. Quantitative data illustrative of test implementation were gathered from routine surveillance forms and key proportions derived. Qualitative data were analyzed thematically. The main challenge to implementing G6PD testing was that only 49.2% (437/888) of eligible patients reached health centers for G6PD testing following malaria diagnosis by community health workers. Factors influencing this included road conditions and long distances to the health center, compounded by the cost of seeking further care and patients' perceptions of vivax malaria and its treatment. 93.9% (790/841) of eligible vivax malaria patients who successfully completed referral (429/434) and directly presented to the health center (360/407) were G6PD tested. Key enabling factors included

**Data Availability Statement:** Qualitative data that are available are de-identified quotations from transcripts. Full transcripts and observation notes are not publicly available as they contain potentially identifiable information. De-identified quantitative data are available as as part of supplementary materials. Raw data is not available due to identifying patient information. However, data are available on reasonable request via emailing to ethics@menzies.edu.au.

**Funding:** This research was supported by a grant from the Australian National Health and Medical Research Council (NHMRC) (1182950). SCS is funded through a Charles Darwin International PhD Scholarships (CDIPS), KT is a CSL Centenary Fellow, and RNP is supported by an NHMRC Investigator Grant (2008501).The funders had no role in study design, data collection and analysis, decision to publish, or preparation of the manuscript.

**Competing interests:** The authors have declared that no competing interests exist.

the test's acceptability among health workers and their understanding of the rationale for testing. Only 36.5% (443/1213) of eligible vivax episodes appropriately received primaquine. 70.5% (165/234) of female patients and all children under 20 kilograms never received primaquine. Our findings suggest that access to radical cure requires robust infrastructure and income security, which would likely improve referral rates to health centers enabling access. Bringing treatment closer to patients, through community health workers and nuanced community engagement, would improve access to curative treatment of vivax malaria.

## Introduction

Malaria remains a major global health burden, causing an estimated 249 million cases in 2022 [1]. Among the five human pathogenic malaria species, *Plasmodium vivax* is the most geographically widespread and has the second highest burden globally with 6.9 to 14.3 million cases yearly [1,2]. *P. vivax* malaria (vivax malaria) is particularly difficult to control because it has a dormant liver stage (hypnozoites) that can reactivate weeks to months after initial infection, causing acute febrile illness (relapses). Relapses are estimated to cause between 60 and 90% of recurrent infections [3].

Effective treatment requires killing both the blood and liver stages of the parasite with a combination of schizontocidal and hypnozoitocidal antimalarial drugs, referred to as radical cure. The 8-aminoquinolone compounds, primaquine and tafenoquine, are the only licensed hypnozoitocidal drugs currently on the market but can cause severe hemolysis in individuals with reduced activity in the Glucose-6-Phosphate Dehydrogenase (G6PD) enzyme. G6PD deficiency is an X-linked enzymopathy present in up to 35% of individuals in vivax endemic regions [4,5]. Until recently the lack of a point-of-care (PoC) rapid diagnostic test (RDT) to diagnose G6PD deficiency and related concerns of drug-induced hemolysis limited the use of primaquine in malaria-endemic areas [6]. However, PoC G6PD tests have now been developed, including qualitative lateral flow rapid diagnostic tests (e.g., CareStart G6PD RDT, Access Bio, USA) and semi-quantitative handheld devices (e.g., STANDARD G6PD test, SD Biosensor, ROK) [7–9]. Eight vivax endemic countries have recently started implementing the STANDARD G6PD test (Biosensor), and several others are preparing to use the test as part of national malaria control treatment strategies to improve radical cure treatment [10].

In Cambodia the burden of malaria has fallen considerably. The number of cases reported yearly has decreased from 113,855 cases in 2004 to 4,021 in 2022, leading the National Center for Parasitology, Entomology and Malaria Control (CNM) to set a target to eliminate malaria within its borders by 2025 [11–13]. The proportion of vivax malaria, however, has increased relative to falciparum malaria, accounting for 89% of all malaria in 2022 [12]. Although *P. vivax* has caused the majority of malaria cases since 2012, it only became a priority for Cambodia's malaria program in 2019 [14].

Primaquine, at a low dose of 3.5mg/kg, has been recommended in the Cambodian National Treatment Guidelines since 2014 as a treatment for vivax malaria in patients with confirmed G6PD normal status, while deficient individuals were not to receive primaquine [15–17]. Despite the recommendation for G6PD normal individuals, patients were not treated with primaquine because of a relatively high prevalence of G6PD deficiency of 8–19% [18–21], a lack of PoC G6PD testing [15], and concerns about severe hemolysis following drug exposure [15].

Once PoC G6PD testing became available, CNM piloted the routine use of the CareStart G6PD RDT in 2019 and the Biosensor in 2020 in Pursat province, during which 57% of vivax and cases with mixed infections (*P. vivax* and *P. falciparum*) received primaquine after testing in a more controlled and supervised setting [22]. When expanded to four additional provinces between November 2019 and December 2020, further challenges to implementing the Biosensor were identified as only 29.6% of vivax and mixed cases were tested and received primaquine during that time period [23]. Lessons learned from this scale-up, including the need to strengthen the referral pathway and enhance monitoring and supervision in high burden areas, were used to tailor strategies to improve coverage prior to the nationwide rollout [23]. In 2021 Cambodia was one of the first countries to implement the Biosensor across provincial and district hospitals and health centers that had previously recorded one or more patients with vivax malaria per month.

Implementation of new diagnostics such as RDTs for use at PoC, can be challenging because of a multitude of factors at macro (political, social, and economic), meso (organizational and institutional), and micro (individual) levels [24]. Among these factors are inadequacies in the supply chain [25–27], training [25,26,28] and emphasis on guidelines [26]; infrastructure [29,30], logistic and financial barriers [25,31]; workload and human resources constraints [26,30]; and preferences for embedded diagnostic processes [26,31]. In addition, patient-provider dynamics impact the implementation of new technologies [26–28,32]; these include understandings of quality of care [32], lack of alternative diagnostics and treatment [27,32] or patient expectations of care [26,28], how health workers perceive socio-economic burden on patients [30], and a lack of coordination and interaction among providers, and between providers and patients [27].

In this study we assess the rollout and implementation of the Biosensor and its role in vivax case management after its countrywide rollout and its first year of routine use in Cambodia. We explore how malaria service providers and recipients experience, perceive, and receive G6PD testing and examine the impact of introducing G6PD tests as a part of radical cure treatment for vivax malaria.

## Methods

### Study overview

In this mixed-methods study, we used a convergence triangulation approach in which qualitative and quantitative data are collected and analyzed separately, but results from both are compared and interpreted together [33,34]. In our study, qualitative data were collected to explore stakeholders' perceptions of G6PD testing and factors influencing its implementation, while quantitative data were gathered to triangulate qualitative findings and further describe the adoption of the test. Qualitative and quantitative data were collected at different times but cover the same time period (January 2021 to March 2023) of analysis.

To guide data collection and analysis, we applied a conceptual framework (Fig 1). The applied framework mainly utilizes Proctor *et al.*'s ' implementation outcomes' (acceptability, adoption, appropriateness, feasibility, fidelity, implementation cost, and sustainability) [35] and also includes expectations of the intervention from Greenhalgh and Russel's framework [36]. Elements of normalization process theory were also integrated into the framework (coherence) to enable a comprehensive picture of the implementation of G6PD testing [37]. Normalization process theory focuses on building a theory to explain how material practices become routinely embedded in social contexts as a result of people working individually and collectively to implement them [37].

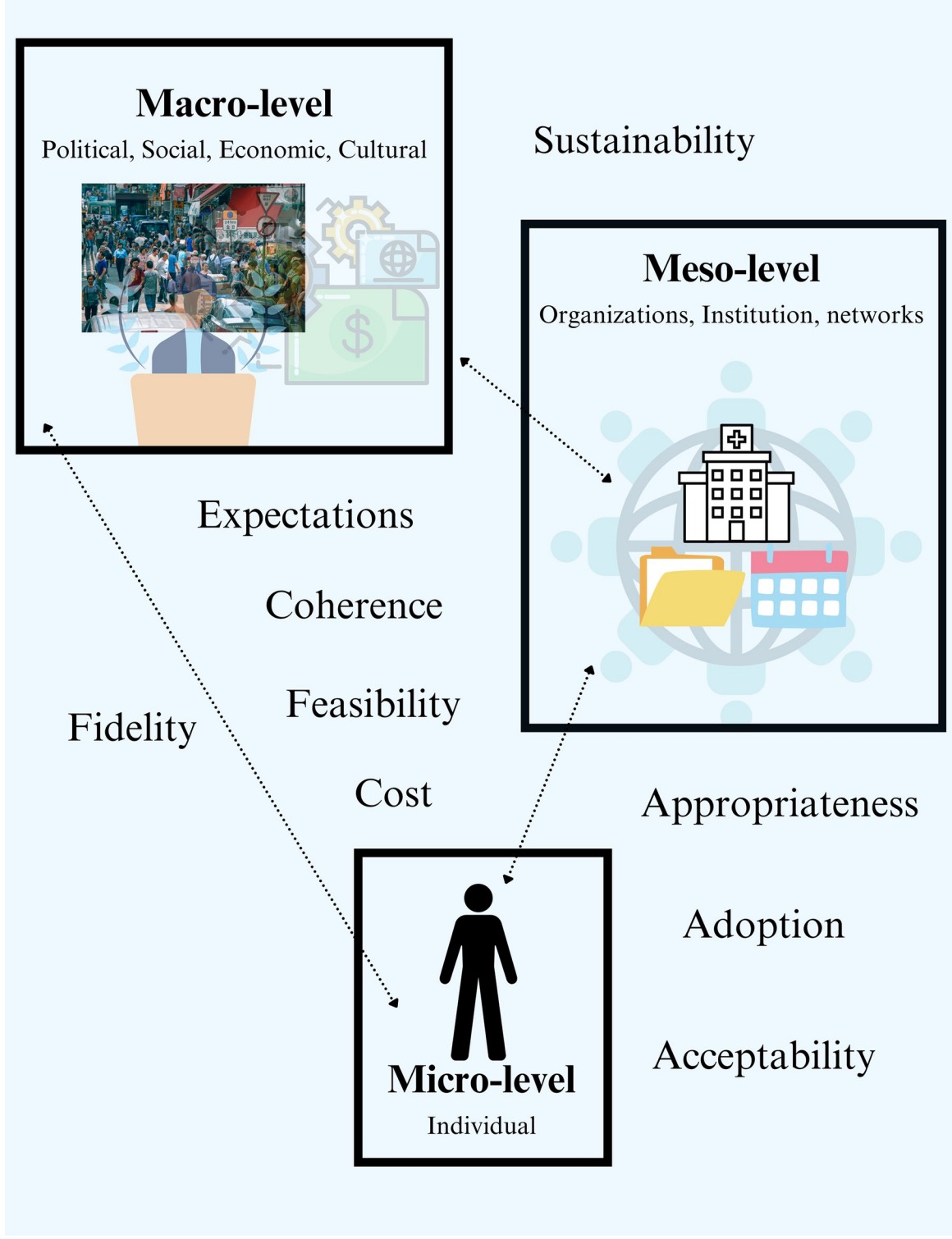

**Fig 1. Conceptual framework developed for data collection and analysis.**

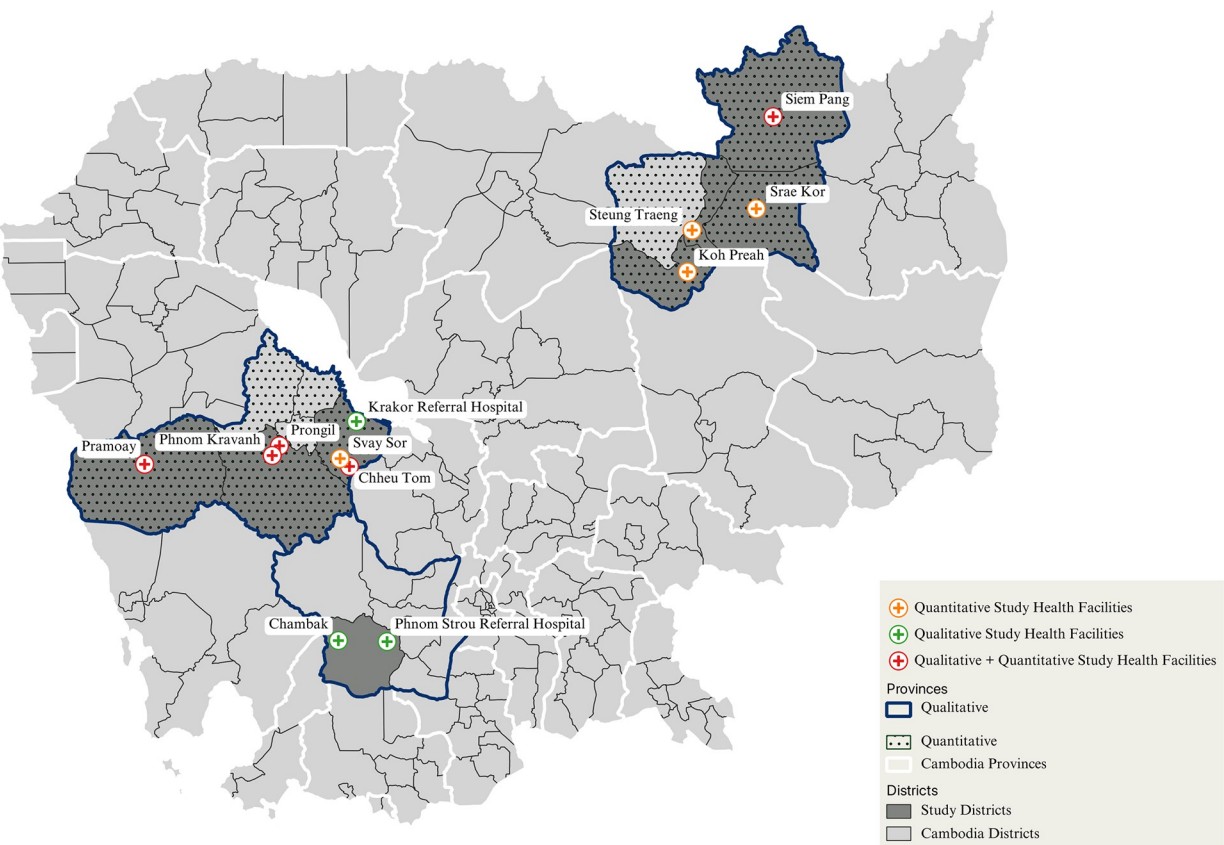

**Fig 2. Map of Cambodia highlighting study sites according to type of data collected.** Map designed and generated using QGIS Lima. Shapefiles for provincial and district boundaries were obtained from the Humanitarian Data Exchange (https://data.humdata.org/dataset/cod-ab-khm) and are licensed under a Creative Commons Attribution 4.0 (CC-BY 4.0) International license.

## Study sites

Data was collected in a total of 12 study sites. At five sites, qualitative and quantitative data were collected; at three sites, only qualitative data were collected; and at four sites, only quantitative data were collected (Fig 2).

For the qualitative strand, study sites included six routine health facilities and/or their catchment areas and two sites where a clinical trial was being conducted to assess the effectiveness of different new radical cure regimens, which required patients to be tested using the Biosensor (EFFORT study, NCT04411836).

The quantitative strand was conducted at three health facilities that provided only routine care and six facilities that provided routine care and care as part of research studies, among them the EFFORT study.

The study sites were chosen to include health facilities with varying malaria burdens providing routine care only, as well as facilities providing routine care and research-related care given the malaria research-saturated context at health facilities in Cambodia.

## Overview of routine vivax case management policy in Cambodia

In 2021 the Biosensor was introduced at health centers, with CNM recommending that patients diagnosed with vivax malaria should be prescribed schizontocidal treatment

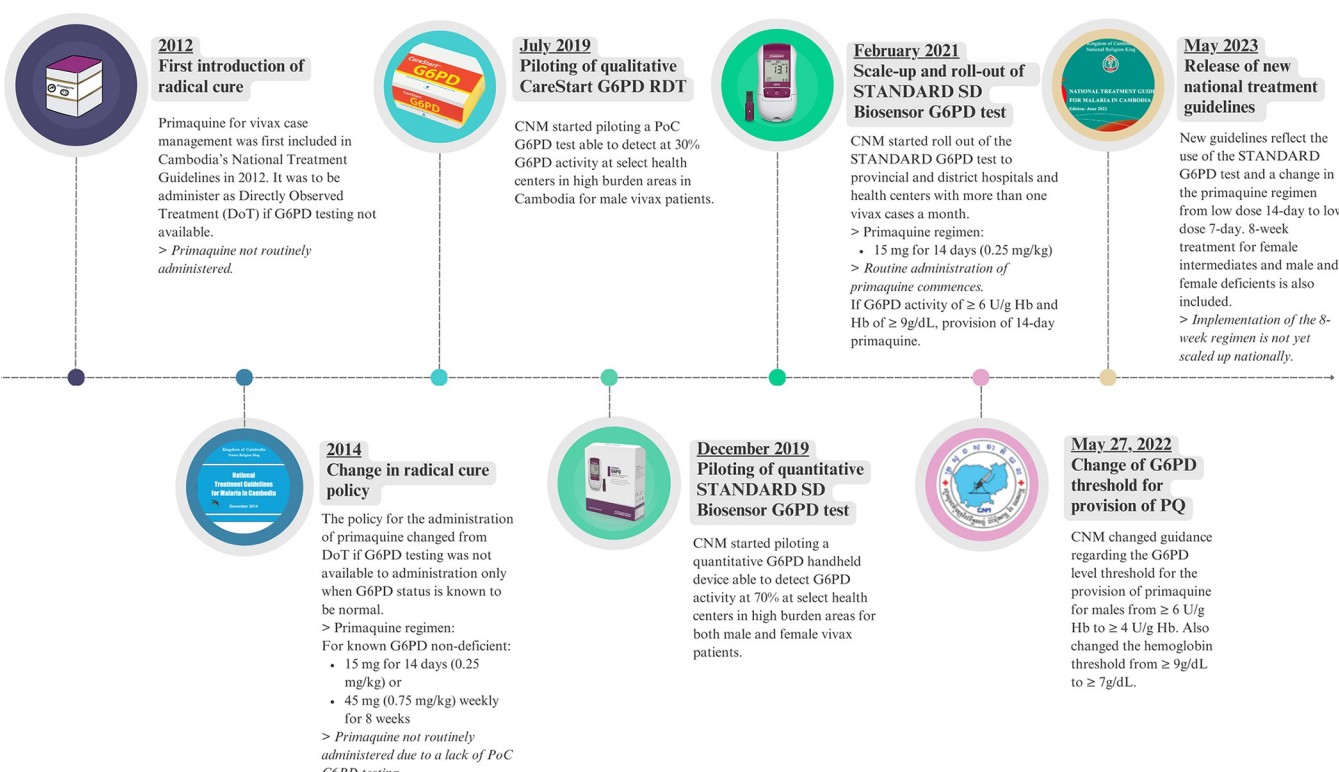

**Fig 3. Timeline of Cambodia's *P. vivax* case management policy and implementation.** Images of Access Bio G6PD RDT and SD Biosensor STANDARD G6PD test sourced from manufacturer websites [39,40].

(artesunate-mefloquine) and tested for G6PD deficiency prior to the administration of primaquine (Fig 3) [38].

Per national guidelines, G6PD status was defined based on quantitative Biosensor test results as normal (≥6U/gHb, corresponding to ≥70% enzyme activity), intermediate (4-6U/gHb, 30–70% activity) or deficient (<4U/gHb, <30% activity). If G6PD test results were ≥ 6U/gHb, patients could be administered primaquine. Starting from June 2022, if G6PD test results for male patients were ≥4U/gHb, they were classified as G6PD normal and were administered primaquine [41]. For females the threshold of 6U/gHb for primaquine administration remained. The hemoglobin level allowing the provision of primaquine also changed in June 2022 from ≥ 9g/dL to ≥7g/dL [41]—the cutoff below which the Biosensor does not generate reliable G6PD readings.

During the study period, the recommended dose of primaquine was 0.25 mg/kg/day for 14 days for G6PD normal patients. Patients who were diagnosed as G6PD intermediate or deficient were recommended to be treated with schizontocidal treatment alone unless referred to research studies.

Routine G6PD testing was only introduced at health centers. Hence, patients diagnosed with vivax malaria by community health workers (CHWs) were treated with schizontocidal drugs in the community but required referral to a health center for testing prior to primaquine administration. Exceptions to referral for testing and treatment with primaquine included patients who weighed less than 20kg and those with contraindications for primaquine (pregnancy and breastfeeding) who were only eligible for schizontocidal treatment [16].

### Qualitative strand

**Sampling frame.**    Between April and November 2022, key informant interviews (KIIs) and focus group discussions (FGDs) were conducted with stakeholders across three provinces, 10 districts, three referral hospitals (two routine care sites and one EFFORT trial recruitment site), and six health center catchment areas (four routine and two EFFORT) (Fig 2). Where required and possible, follow-up interviews and FGDs were conducted in 2022 and between March and April 2023.

In line with normalization process theory, perspectives from multiple stakeholders were obtained across sites with varying vivax malaria burden. Respondents included current and former patients, CHWs, health facility staff—including doctors, nurses, and midwives—provincial and district health and malaria officials, EFFORT trial staff, CNM officials, and implementation partners. In selecting participants, purposive and convenient sampling was undertaken based on experience, demographic characteristics, study sites, and availability of key informants. Recruitment and enrolment of individuals previously diagnosed with vivax malaria involved convenient and snowball sampling (peer referral) and was coordinated with health facility staff at the respective sites as well as CHWs and community leaders.

Observations of G6PD testing procedures and workflow were conducted in two of the routine health center catchment areas and at one of the EFFORT study sites.

**Data collection.**    Prior to commencing the KIIs and FGDs, semi-structured discussion guides (Appendix A in S1 Text) were developed in English based on the conceptual framework (Fig 1). The discussion guides were translated into Khmer, and the meaning of questions was crosschecked through back translation and workshops with facilitators (KC, PC). The guides were then tested in a pilot study. The semi-structured approach to the KIIs and FGDs allowed for an iterative process.

Written informed consent was obtained from all participants prior to commencing the KIIs and FGDs. All but one KII with policymakers were conducted in English, while all others were in Khmer. The KIIs and FGDs were recorded, and English recordings were transcribed, while those in Khmer were transcribed and translated into English by three translators. The study team reviewed the translated transcripts to ensure clarity of meaning.

Direct observation of G6PD testing procedures and workflow, as well as general health facility workflow, was also conducted. Observations were conducted by at least two members of the study team, one of whom spoke Khmer. Observations were non-participant and overt. Observations were recorded on paper using a notetaking grid for which general themes were pre-determined and adjusted based on initial observations. Pictures were taken to supplement notetaking. Notes from each observer were electronically compiled, and pictures were incorporated into observation records. Informed written consent for the observation, including picture-taking, was obtained from healthcare workers by study staff upon arriving at the study site and from patients by health center staff when patients presented to the outpatient department with a fever.

**Data analysis.**    Transcripts were reviewed, and a codebook was developed. NVivo 12 (Lumivero, United States) was used for coding, which entailed a combination of inductive and deductive approaches. Initially, transcripts were read without any preconceived notion of themes and initial codes identified. In an iterative process the transcripts were read again, and a final codebook was generated with the initial codes and those generated using the key components of the framework (Fig 1). Transcripts were then coded, and emerging themes were analyzed through memo writing. Notes from both observers were collated, and insights from all observations were synthesized with findings from KIIs and FGDs and are reported based on the themes emergent from KII and FGD analysis, identifying respondent type where

**Table 1. Demographic characteristics of KII and FGD respondents.**

| Respondent Type Characteristic | Patients (pp) | Patient spouses (ps) | CHWs (chw) | Health Center Staff (hc) | Referral Hospital Staff (rh) | Malaria Officials (cnm, po, do) | Partners (cnmp) | EFFORT Trial (et, etl, etd) | Overall |
|---|---|---|---|---|---|---|---|---|---|
| Females | 6 | 4 | 28 | 4 | 5 | 2 | 0 | 8 | 57 |
| Males | 29 | 0 | 21 | 8 | 7 | 10 | 3 | 7 | 85 |
| Median Age (IQR, range) | 30 (20–39, 18–68) | 31.5 (23.5–31.5, 20–39) | 50 (39–57, 23–71) | 45 (33–54, 26–55) | 31 (29–52, 26–57) | 50 (31–50, 28–70) | 42 (34–50, 34–50) | 31 (28–34, 24–57) | 36 (29–51, 18–71) |
| Unknown Age | 0 | 0 | 0 | 1 | 0 | 3 | 1 | 0 | 0 |
| Villages | 32 | 4 | 46 | 10 | 10 | 10 | 3 | 10 | 108 |
| Occupations | Farmer (19), Forest product gatherer (7), Daily laborer (2), Forestry administration (2), Teacher (1), Environmental officer (1), Construction worker (1), School director (1), Environmental NGO staff (1) | Farmer (2), Grocery seller (2) | Farmer (42), Grocery seller (4), housewife (1), Online seller (1), Unknown (1) | Malaria Point Person (4), Doctor (1), Nurse (2), Midwife (2) Laboratory (2) Administrative (1) | Doctor (1), Nurse (7), Laboratory (2), Administrative (2) | CNM (3), PHD Director (2), PMS (2) OD Director (2) ODMS (3) | Central level partner (1), District level partner (2) | Doctor (4), Nurse (4), Laboratory (4), Administrative (3) | 25 |
| Total | 35 | 4 | 49 | 12 | 12 | 12 | 3 | 15 | 142 |

IQR = Interquartile Range, PHD = Provincial Health Department, PMS = Provincial Malaria Supervisor, OD = Operational District, ODMS = Operational District Malaria Supervisor.

possible using unique codes (Table 1). During the analysis, the study framework was expanded to include device-level factors in addition to macro, meso, and micro level factors.

## Quantitative strand

**Sampling frame and data collection.** Between March (16/03/2023) and April (29/04/2023), quantitative data routinely collected as part of malaria surveillance was gathered from nine health centers across seven districts in two provinces (Fig 2). Study health centers included Phnom Kravanh, Pramoay, Chheu Tom, Prongil, and Svay Sor in Pursat Province; and Siem Pang, Koh Preah, Srae Kor, and Steung Traeng in Steung Traeng Province (Fig 2).

The data collected included relevant variables for all individuals presenting for care between January 2021 (01/01/2021) and March 2023 (31/03/2023). The variables were gathered from national routine surveillance forms for individuals presenting for care within the nine health center catchment areas. Data on the frequency of G6PD testing, G6PD test results (numeric reading), and treatment provided were collected (Table A in S1 Text). Information was captured on individuals presenting directly to health centers, as well as those presenting to CHWs —which include village malaria workers (VMWs) and Mobile Malaria Workers (MMWs). These individuals included patients with vivax malaria who were referred to local research studies.

With the support of CNM implementing partners, data from four sources were gathered from health centers and CHW; these included health center outpatient department and vivax malaria logbooks and CHW case registries and follow-up forms. The study team then entered information for each malaria case recorded on these forms into REDCap hosted at Menzies School of Health Research, Darwin, Australia [42,43]. Duplicate cases recorded by both CHWs and health centers were only entered once after being identified by matching patient

information and date of patient presentation. Data entered into REDCap were exported into Microsoft Excel for cleaning and preliminary analysis, which consisted of identifying outliers or incongruous data, which were then validated by the study team. Data collected was also crossed checked with available aggregate health center malaria surveillance data (Table B in S1 Text).

Informed consent from patients was not obtained given that the use of routine data was approved by respective ethics committees without the need for further individual consent.

**Data analysis.** A 'vivax episode' was defined as individual vivax presentations that were diagnosed at health centers or by CHWs during the study period. A 'vivax patient' was defined as the same individual diagnosed with vivax malaria who presented at the health center or to a CHW with one or more episodes. Recurrent vivax episodes were identified by matching patients by their name, age, sex, village name, and contact number. A vivax episode was considered recurrent if the record had a similar name, a combination of two or more of the other elements matched, and did not occur within a week of the previous episode. In a validation step, another member of the study team reviewed the identified recurrent vivax episodes and the matching process. The study team subsequently assigned a unique study ID for all patients. Data cleaning and manipulation to identify recurrent cases without de-identification was done by two investigators only. Data was then de-identified for further analysis.

For our analysis, G6PD status was defined according to the recorded G6PD numeric test results and thresholds used in guidelines at the time of diagnosis. Administration of appropriate treatment was defined as treatment according to guidelines at that timepoint (Fig 3). The incidence rates of recurrent malaria were calculated from the number of recurrent episodes per patient for the duration of the study period (27 months). Chi-squared tests were used to assess statistically significant differences in proportions. The analysis was conducted in Stata 18.0 (StataCorp, United States).

## Ethics

Ethical approval was obtained from the Cambodian National Ethics Board (# 118), the Institutional Review Board of the Menzies School of Health Research (HREC 2020–3694), the Charles Darwin University Human Research Ethics Committee (H22047), and the Oxford Tropical Research Ethics Committee (Ref. 39–20).

## Results

### Participant characteristics

**Qualitative strand.** A total of 67 KIIs and 19 FGDs were conducted in two rounds of data collection in 23 villages across 10 districts in Cambodia (Table C in S1 Text). A total of 142 respondents participated, 22 (15.5%) of whom also took part in follow-up KIIs (n = 6) and FGDs (n = 5). Respondents included 35 patients with vivax malaria (19 in the clinical trial and 16 in routine care) [pp], four spouses of patients present during routine care [ps], 49 CHWs (40 VMWs and 9 MMWs) [chw], 12 health center staff [hc], 12 referral hospital staff [rh], five district malaria officials [do], four provincial malaria officials [po], three policymakers [cmn], three implementing partners [cmnp], and 15 clinical trial staff [et, etl, etd]. Of the 35 patients, the mean age was 32 (total range 18–68); 29 (82.9%) were males, and 27 (77.1%) were subsistence farmers or forest-goers (Table 1 and Table D in S1 Text). Patients came from 32 different villages.

Study team members conducted 34 observations of febrile patient visits to healthcare providers who were using the Biosensor as part of either routine care or the clinical trial.

Observations were done over a total of four hours and 33 minutes across three observers.

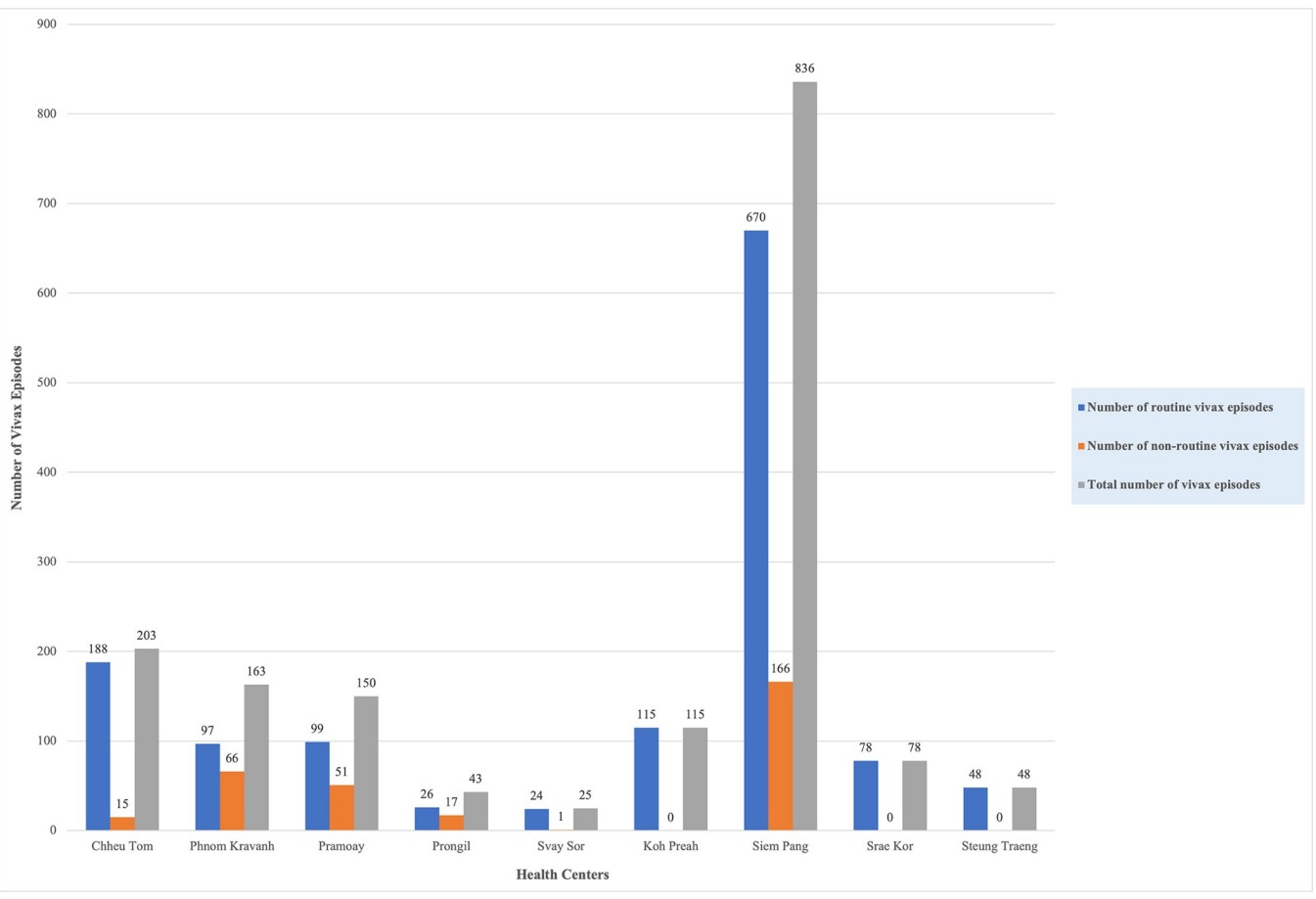

**Fig 4.** *Plasmodium vivax* **episodes for which routine and non-routine care was received from 9 selected health center catchment areas in Cambodia between January 2021 and March 2023.** Routine refers to those who presented to CHWs or health centers and received routine testing and treatment. Non-routine refers to those who presented to CHWs or health centers and were referred to studies where they received testing and treatment according to study protocols.

**Quantitative strand.** Overall, 1,661 vivax episodes were recorded from 1,334 patients living in 184 villages within the catchment areas surrounding nine health centers (≥88% agreement with surveillance data, Table B in S1 Text). The first episode included in the analysis was recorded on January 5, 2021, and the last on March 31, 2023. The health center in Siem Pang contributed more than half of the recorded episodes (836; 50.3%), whereas 203 (12.2%) episodes were recorded from Chheu Tom and only 25 (1.5%) were from Svay Sor (Fig 4 and Table E in S1 Text). Of the 1,334 patients with at least one record of vivax malaria, 1,100 (82.5%) were male, and 771 (57.8%) were under 25 years old.

## Adoption of the Biosensor and vivax case management policy

Results on the adoption of the Biosensor and vivax case management policy are presented in the subsequent three sections according to the flow of vivax case management: (1) referral of vivax cases diagnosed at the community level by CHWs to health centers, (2) G6PD testing at the health center, and (3) treatment after G6PD testing at the health center.

**Referral from community health workers to health centers.** Out of 1,661 recorded episodes, a total of 1,183 (71.2%) of the vivax episodes were diagnosed in the community, requiring patient referral to a health center for G6PD testing and subsequent primaquine treatment.

This proportion varied from 0.0% to 89.4% across the health centers (Table F in S1 Text). For 11.7% (138/1183) of the episodes, patients were referred to research studies; for 13.3% (157/1183) of the episodes, patients were not eligible for testing and treatment with primaquine (Fig 5). Hence, for 888 (75.1%) episodes, patients should have been referred for testing and treatment, but only for 437 (49.2%) episodes were patients successfully referred and presented at the health center for G6PD testing and treatment. The proportion of episodes for which referral was successful varied across health center catchment areas, ranging from 0% (0/9) successful referrals in Srae Kor to 10.0% (4/40) in Koh Preah and 30.9% (231/747) in Siem Pang. Conversely, nearly all patients were referred to the health center in Prongil (23/24) and Svay Sor (22/22) (Table F in S1 Text).

In the qualitative analysis, respondents mentioned that the travel required from the community to the health center for patients to be tested for G6PD deficiency was a significant barrier to recommended case management (cnm1-3, cnmp3, hc5,8). Factors influencing travel to the health center were macro-level or structural, meso-level (organizations and network), and micro-level (individual) factors (Fig 6). Two main contributors were identified: the availability and cost of transportation (cnm2-3, po3-4, chw13, et1, et5) [micro and macro] as well as the distance from patients' village or forest fringe to the health center (cnm3, po4, hc5, chw19,24, pp3,6,17,20,23) [macro]. These were compounded by difficult road conditions (cnm2, chw13, pp21,23) [macro], opportunity costs of traveling to the health center (cnm2, chw23,26) [micro and macro], patients' previous treatment experience (e.g., reinfection despite radical cure treatment) (po4, rh5, rh [fgd7], pp24) [micro], and patient's perception and understanding of the disease and its treatment (cnm2-3) [micro], including perception of vivax as benign (cnm2), not requiring treatment (cnm2), the length of perceived standard treatment (hc4,9, chw1,7,26), fear of side effects (cnm2, po4, hc3, rh8), and impact of treatment on work (cnm2, rh8 hc3,7, chw26, pp12) (Fig 6). Such compounding was illustrated by minimal changes in referral rates in the quantitative dataset even after CNM started to provide incentives to CHWs to travel with patients to health centers to address challenges of transportation cost and distance (cnm3, po4).

Since the implementation of incentives to support referral in September 2022, a CNM official indicated that the rates of successful referral increased, though not as much as the national program anticipated. The quantitative data suggested a limited effect. The percentage of patients diagnosed by CHWs reaching health centers in the study sample increased slightly from 46.6% (89/191) recorded between September 2021 and March 2022 before the incentive scheme was initiated to 50.5% (103/204) recorded between September 2022 and March 2023 after the start of the scheme (p = 0.649).

Hence, difficulties in referral were more likely related to difficult road conditions when traveling to the health center (cnm2, chw13, pp21,23), e.g., in flooding or muddy conditions, and the opportunity cost involved in traveling to the health center (cnm2, chw23,26), e.g., needing to collect the day's yield for their families. As emphasized by CHWs, for some patients, not traveling to the health center was not about not wanting to receive complete treatment but that having an income to pay for food and schooling of their children was a priority: "*Yeah, they want to have the complete treatment* [which they only receive if they are G6PD tested], *but their income is more important for the present time*" (chw23).

CHWs differentiated between patients who lived in a village and mobile workers or forest-goers. The opportunity cost for mobile workers or forest goers was greater than for village-based patients, as they needed to travel farther and spend more time away from work (hc5, chw19,23,26,35).

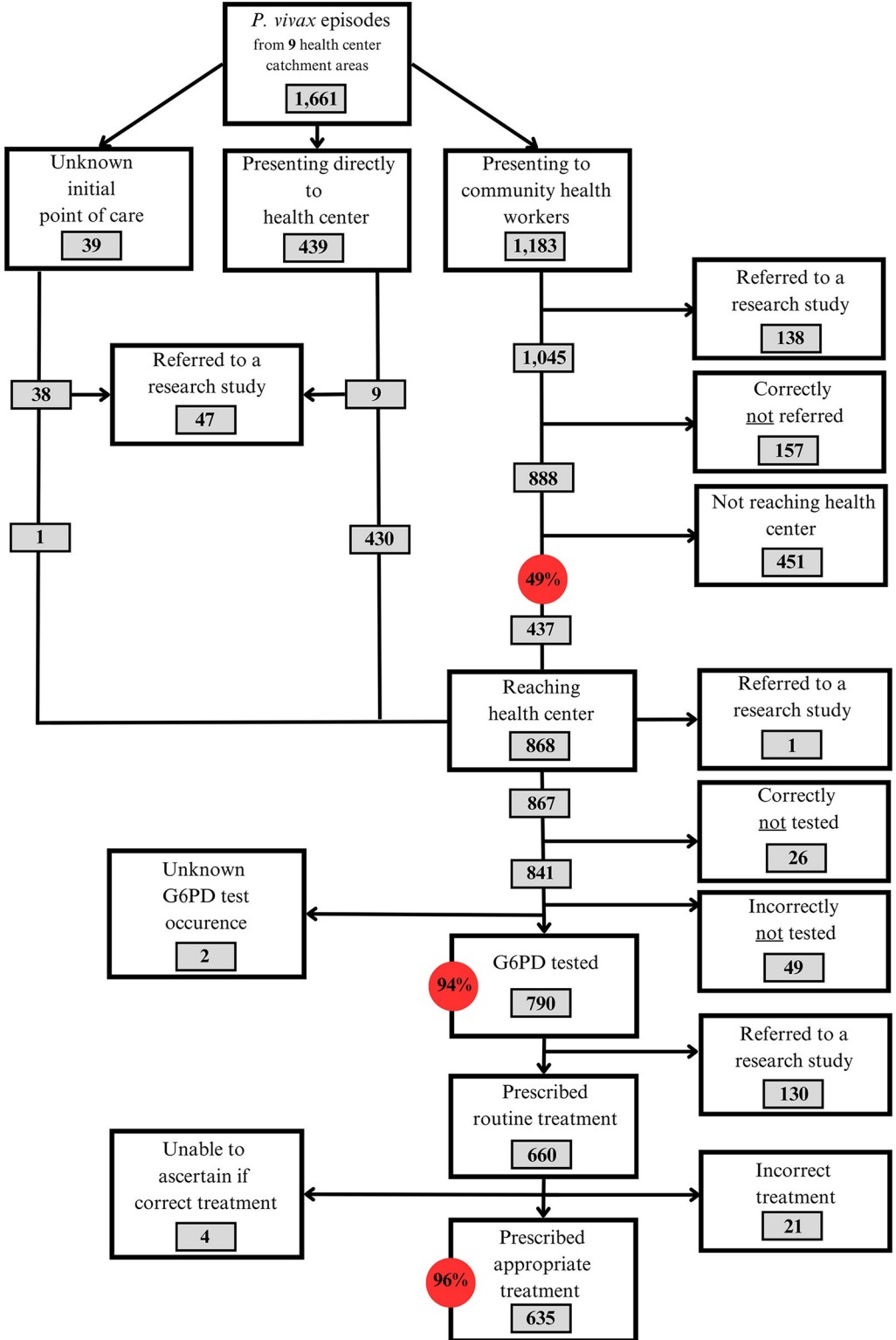

**Fig 5.** *Plasmodium vivax* **case management pathway for patients in 9 health center catchment areas in Cambodia between January 2021 and March 2023.** Red circles highlight key percentage indicators in the implementation of *P. vivax* case management.

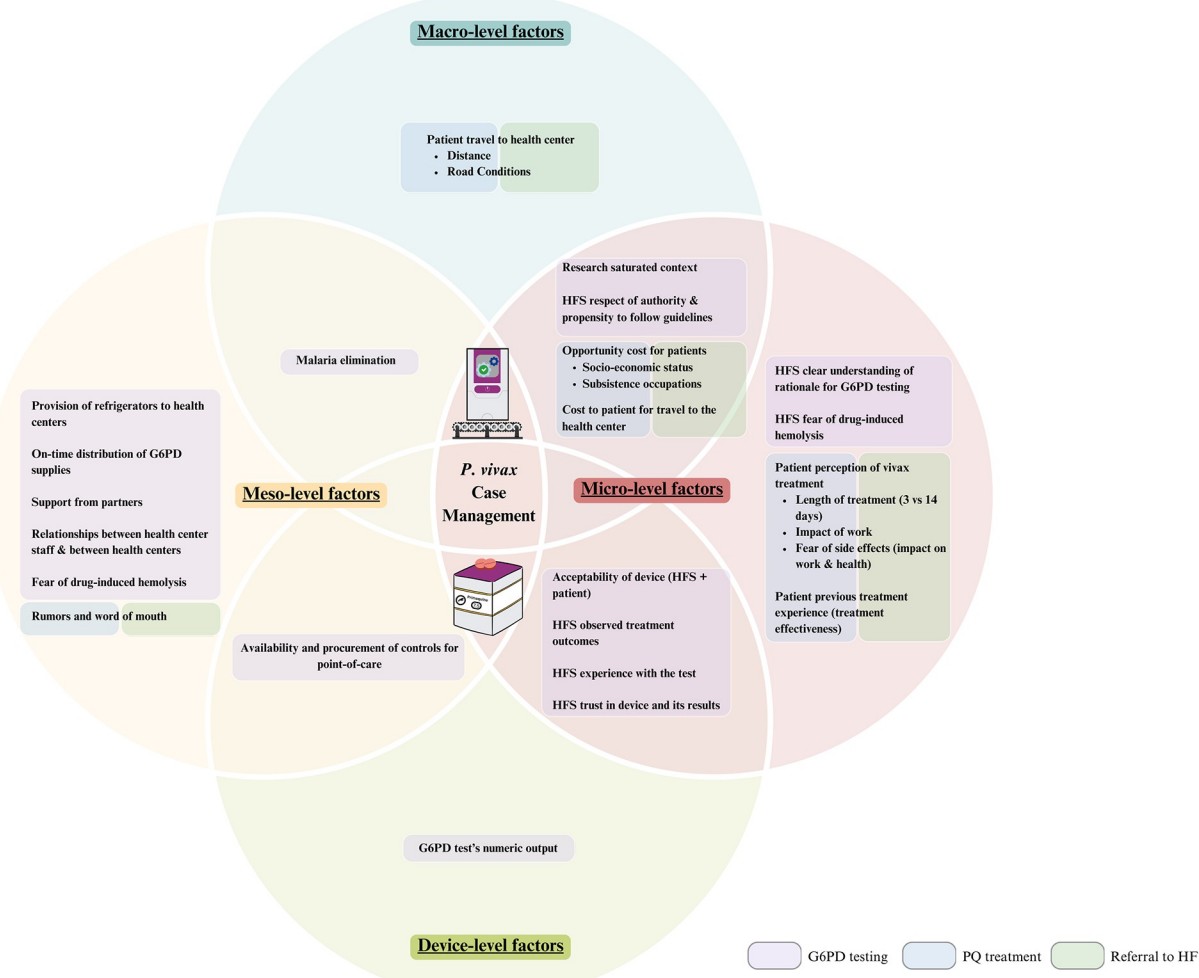

**Fig 6. Factors affecting the implementation and adoption of the Biosensor and *P. vivax* case management.** HFS stands for health facility staff (includes health center and referral hospital staff), and HF stands for health facility.

"*Yes, they don't come back to the health center because they live far from the villages, and if we ask them to come back to the village for the treatment* [which is only available after testing]*, they will not come because they can't leave their personal belongings in the forest, [. . .], they will not come because they earn nothing yet.*"–chw26

The difference in seeking further care between mobile populations and those residing in the village explained some of the geographic variation seen in the quantitative data (Table F in S1 Text).

Patients' perception of disease severity and understanding of vivax treatment also influenced adherence to referral. Per a CNM official, for some patients, malaria, especially vivax, is an illness that they can live with and a disease for which interrupting their work to earn their livelihoods might not be worth it (cnm2). The perception of vivax as a less serious form of malaria than falciparum malaria was also a view maintained by some CHWs (chw17,27).

Beyond how patients perceived the disease itself, perceptions around vivax treatment also contributed to seeking further care at health centers. The 3-day schizontocidal malaria

treatment was available with CHWs, and patients understood malaria treatment as a 3-day regimen, rather than a 3-day regimen in addition to 14 days of primaquine (hc3, chw1,7,26). This understanding hindered referral because patients thought that they had already received the full treatment at the community level (hc3, chw1,7,26).

> *"They might think that as they saw their friends taking medicine there for three days, they can recover from malaria, so why do they need to go that far to the health center [. . .] and they don't really understand the treatment process. The patients think that after taking medicine for three days is enough for them, they can recover."* –hc3

Some patients also feared side effects and how those might affect their work or health and decided against referral for G6PD testing for treatment (cnm2, po4, hc3, rh8). According to malaria officials and implementation partners, referral hospital staff, and patients, fears were often based on "rumors" or word of mouth that spread through communities (poc4, cnmp2, rh [fgd7], pp3). Despite CHWs and patients who had received primaquine highlighting that side effects were experienced more during the first three days of malaria treatment (pp22,24,25,27,30) or associating side effects with artesunate-mefloquine (chw1,2,8,9, pp20,31), difficult experiences or negative associations with the longer treatment existed among some patients (pp3,20):

> "*Before I thought about the 14 days of medicine, I might stay overnight at the hospital or something. I have never tried the medicine, but I heard a lot of patients say that it was so difficult [side effects] for the 14 days of medicine [. . .]*–pp3

Another aspect of treatment perception and previous treatment experience limiting referral was reinfection after previous treatment with primaquine (po4, rh5, rh [fgd7]). Patients noticed and questioned the recurrence of vivax malaria after the prescription of radical cure (pp24, ps2). Explaining such recurrences was a challenge for healthcare providers and meant that some patients might lose trust in primaquine treatment and the health workers administering it (ps24, rh5).

> "*q: What about the 14-day complete treatment? Do you have confidence in it?a: I believe it, but the important thing is that when I go to the forest, I am afraid of malaria again. The previous time I took medicine for half a month, I still had malaria the last time. I told the doctor **that I had malaria twice and had received full treatment but that I had not been completely cured**. He tried to advise me to take the medicine, and I tried to follow him. At that time, after taking the medicine for a year, I got malaria again.*"–pp24

Though referral hospital staff reported one instance wherein a patient did not travel to the health center for complete treatment because "*he thought that we [would] have him stay at the hospital and we take a lot of blood*" (rh [fgd7]), the Biosensor itself is not a direct barrier to referral. Many patients were not aware of the testing and diagnostic process (pp1,2,6,8–10,12,15–17,27,30); and for those who were aware of the Biosensor as an additional test, they perceived it as a malaria test and/or a confirmatory test rather than an additional screening test (pp4,7,11,18, 20,21,24,29, ps3). Patients describing their experiences regarding G6PD testing, not knowing or understanding its purpose, expressed their trust in healthcare providers and the health system despite the discomfort of additional testing (pp28,31). Specifically, if any procedure or test was perceived to be done in the interest of improving their health, patients trust healthcare providers to do what is right for them.

Observations indicated that the lack of awareness of the Biosensor among patients could be a result of limited counseling about G6PD testing. We observed limited communication between patients and health center staff performing the testing and providing the treatment—this being true for both the clinical trial and routine settings. Counseling was provided towards better treatment adherence but to a lesser extent about the interpretation of Biosensor test results and the rationale of conducting the diagnostic itself.

**G6PD testing at health center.** Of the 841 episodes in which patients were eligible for testing and presented to a health center either directly (n = 407) or through successful referral from the community (n = 434), 790 (93.9%) were tested for G6PD deficiency (Fig 5), while 49 (5.8%) were not tested despite being eligible. The occurrence of G6PD testing was unknown for two episodes (0.2%).

The qualitative findings revealed how meso-level, micro-level, and device-specific factors within the context of macro-level factors allowed the Biosensor to be adopted well by health center staff (Figs 1 and 6). Meso-level factors at the central level that supported the adoption included the provision of refrigerators to health centers to maintain appropriate temperatures for test supplies (cnm2,3, cnmp1-3), the regular supply of G6PD test strips (po4, hc9, cmnp1), the procurement of control reagents for quality assurance at the PoC (cnm1,2, po4, cmnp2), and the facilitation of alternative distribution mechanisms for test supplies outside the standard quarterly Ministry of Health supply chain (cnm2).

A meso-level factor exhibited by health centers was the coordination among health center staff who supported each other in using the test (hc5,6) and overcoming stockouts and device malfunctions (hc9).

Micro-level factors contributing to health center staff adopting the test included the acceptability of the device (hc2-9), a clear understanding of the objective of testing (hc1-9), a propensity to follow guidelines (hc1,5,7,9) stemming from a macro-level respect for authority, a level of comfort in conducting relatively complex testing procedures within a research saturated context (hc1,3,5,9), and the fear of drug-induced hemolysis in vivax patients with unknown G6PD activity (hc1,3,4).

The acceptability of the Biosensor among policymakers and health workers lay with its quantitative numeric output as well as the availability of control reagents for quality control purposes that could be run during supervision or at health centers. Policymakers, as well as health workers, found confidence in the device's numeric output allowing for clear interpretation of test results (cnm1,2, po2-4, do1, cnmp2,3, rh4, rh [fgd7], hc5, etd2,3), especially compared to the previously piloted qualitative CareStart RDT; the availability of controls allowing quality assurance also facilitated this sense of confidence in the device (cmn1,2, poc4, do1, hc1,5,9). These two elements, in addition to health center staff observing positive treatment outcomes for their patients (do5, hc6, etd1) and a general trust in the government and national program (hc1,5,7,9, etd1, etd2), generated a level of trust in the device and its results that enabled its use.

Health center workers accepted the Biosensor despite some challenges with the test and testing process. Respondents with varying degrees of experience at health centers, referral hospitals, and trial laboratory staff reported challenges in using the test correctly, especially when first introduced to the Biosensor (cmn1, po4, rh5, rh [fgd7] hc1-3,5–7,9, chw3, cnmp1,2, et3,5, etl1-4, etd2) (Table G in S1 Text). Among other difficulties, they highlighted the complexity of the steps involved as compared to a standard RDT, e.g., malaria RDT, or glucose tests (cnm1,2, rh 8,9, rh [fgd7], etd2, et1,6,7, etl2,4). A health center staff described it as, "*It's a little job, but it's a lot of stories. I mean that there are a lot of possible errors, like collecting the blood sample*" (hc12). After some time using the test, support during monitoring and supervision, and learning about the tests' intricacies, there was a general impression that health center staffs' ability

to perform the test correctly improved (cmn1, cnmp3, po4, etl2). In observing users of varied training and experience, difficulties in the G6PD testing process and its numerous "stories" were apparent—though the level of training has an impact on proficiency and fluidity in conducting the test. These difficulties notwithstanding, health center staff were appreciative of the tests' overall ease of use (hc1-9).

Contributing to the adoption of the Biosensor was also a clear understanding among health workers of the purpose of the G6PD test as a means to provide radical cure safely—a purpose which is different from the majority of other PoC tests used to provide a disease diagnosis (rh1-5, hc1-9, etd1-3).

> "*I think it is a good machine because it is so small, but we can test their enzymes so we can know if they can take medicine or can't take medicine based on the results of their enzymes. When we didn't have that machine, it was so difficult, and we were not allowed to provide treatment, and we didn't have the complete treatment medicine.*"–hc3

Within a macro-research saturated context and culture and facilitating adoption, many health center staff in high-burden areas had been exposed to studies and operational research that had included G6PD diagnostics and laboratory tests, as well as generally complex study procedures (hc1,5,9). Through a pilot study of the Biosensor, health center staff also had pre-exposure to the test itself prior to the rollout and had extensive attention and support from partners for any challenges faced.

While fear of hemolysis without G6PD testing was institutionalized (cmn2), when discussing side effects, health center staff were more focused on patients' responses to treatment side effects than the actual health risk of drug-induced hemolysis (hc2, hc5-12). That said, referral hospital staff, who see fewer malaria cases and do not frequently perform the G6PD test nor administer treatment, did show fear that primaquine could cause a negative health impact on patients (etd3, rhl2).

> "*We use only one dose [of primaquine]. We use one dose for the follow-up. We're afraid of it. After that we used G6PD. It's viral [primaquine can be used widely]. Before using G6PD, we practiced it. [. . .]. We did follow-up based on the side effects of primaquine. We are afraid of the danger.*"–etd3

The small proportion of patients in the quantitative dataset who were not tested was likely grounded in health center staff giving patients the choice of whether to receive the 14-day primaquine treatment in addition to the 3-day schizontocidal treatment (po4, hc4 etd1). Hence, for patients who chose not to receive primaquine, health center staff did not perform G6PD testing. Other reasons for non-adherence to G6PD testing guidelines were the absence of the health center staff responsible for G6PD testing (do1) and stockouts of supplies specifically affecting Siem Pang, which had a high caseload. Non-adherence to G6PD testing guidelines also occurred at a referral hospital when a doctor's fear of prescribing primaquine as part of vivax case management prevented them from using the G6PD test (rhl2).

**Prescribing primaquine after G6PD testing.**    Among the 660 episodes in which patients were tested for G6PD deficiency, 635 (96.2%) were treated appropriately based on test results (Fig 5, Table H in S1 Text); 443 (67.9%) were found eligible for treatment and received primaquine. In four cases the measured G6PD result was not recorded, therefore, appropriate treatment could not be determined.

The qualitative data suggests that the same factors that led health center staff to perform G6PD testing also ensured that they prescribed the appropriate treatment. These factors

included trust in the G6PD test result, fear of primaquine-induced hemolysis without testing, the observed impact of radical cure, and a sense of responsibility to follow guidelines (hc1,4,5,7,9) (Fig 6).

Of the 21 episodes in patients who were treated incorrectly, 13 (62.0%) were not treated with primaquine despite testing G6PD normal, and eight received primaquine treatment despite not fulfilling the criteria for treatment. Of the eight incorrectly receiving primaquine, in four (50.0%) episodes the patient had hemoglobin level below the threshold for treatment, in three (37.5%) episodes the patient was diagnosed as G6PD deficient, and in one (12.5%) episode a female patient with intermediate G6PD activity was treated with primaquine. No data were available on clinical outcomes of the eight patients incorrectly receiving primaquine.

According to the qualitative data, reasons for health center staff not prescribing primaquine despite normal G6PD test results included both health center and patient factors influenced by structural, macro-level conditions. A health center staff described assessing patients' overall health condition (e.g., pallor) after testing and deciding not to provide primaquine, fearing that patients' condition would worsen (hc1). A malaria official and healthcare provider also mentioned (po4, rh3) occasions when patients were tested for G6PD, but after they were tested, the patients decided that they did not want the radical cure treatment because it "*wastes their time to earn money or to work*" (rh3).

The thresholds for different G6PD statuses, as well as that for hemoglobin, have changed in Cambodia during the study period without training for all health center staff, resulting in some confusion regarding G6PD test result thresholds that could have contributed to some of the incorrect treatment provision (hc2,7,10).

## Impact on the provision of primaquine treatment

Based on the qualitative data, the expectation for the implementation of the Biosensor was that G6PD testing would allow patients with vivax malaria to safely access radical cure and reduce vivax burden (cmn2, po1,4, do1, hc7, cnmp1). Despite such expectations, quantitative indicators suggest that the implementation of the Biosensor, including referral policy, has not yet facilitated access to radical cure for most vivax patients. Of the 1,661 recorded vivax episodes, 448 (27.0%) were not eligible for primaquine before G6PD testing (pregnant, breastfeeding, under 20kg) or after testing (G6PD intermediate/deficient and low Hb). Of the eligible 1,213 vivax episodes, only 443 (36.5%) were appropriately treated with primaquine after routine G6PD testing. For 271 episodes, patients were treated with different radical cure regimens in a research context (Fig 5).

## Impact on treatment outcomes

A total of 1,661 vivax episodes were recorded in 1,334 patients, of whom 1,114 (83.5%) had only one recorded episode of vivax malaria, and the remaining 220 patients (16.5%) experienced two or more episodes. One patient had seven recurrences (Fig 7). The median time between episodes was 75 days (IQR: 53–134, range: 9–611), with 69.3% of recurrences occurring within the first 120 days (Fig A in S1 Text).

Of the 1,334 patients, 595 (44.6%) received primaquine for all of their episodes (range 1–4 episodes) and 661 (49.6%) for none of their episodes (range 1–8 episodes). 78 (5.8%) patients with more than one episode received treatment only for some of them. Of the 661 patients never treated with primaquine, 81.5% (539) were diagnosed by CHWs compared to 53.6% (319) of those who always received primaquine (p<0.001) (Table 2); and 16.0% (106) of those never treated with primaquine weighed less than 20 kg and therefore were not eligible for primaquine compared to none among those always having received primaquine (p<0.001).

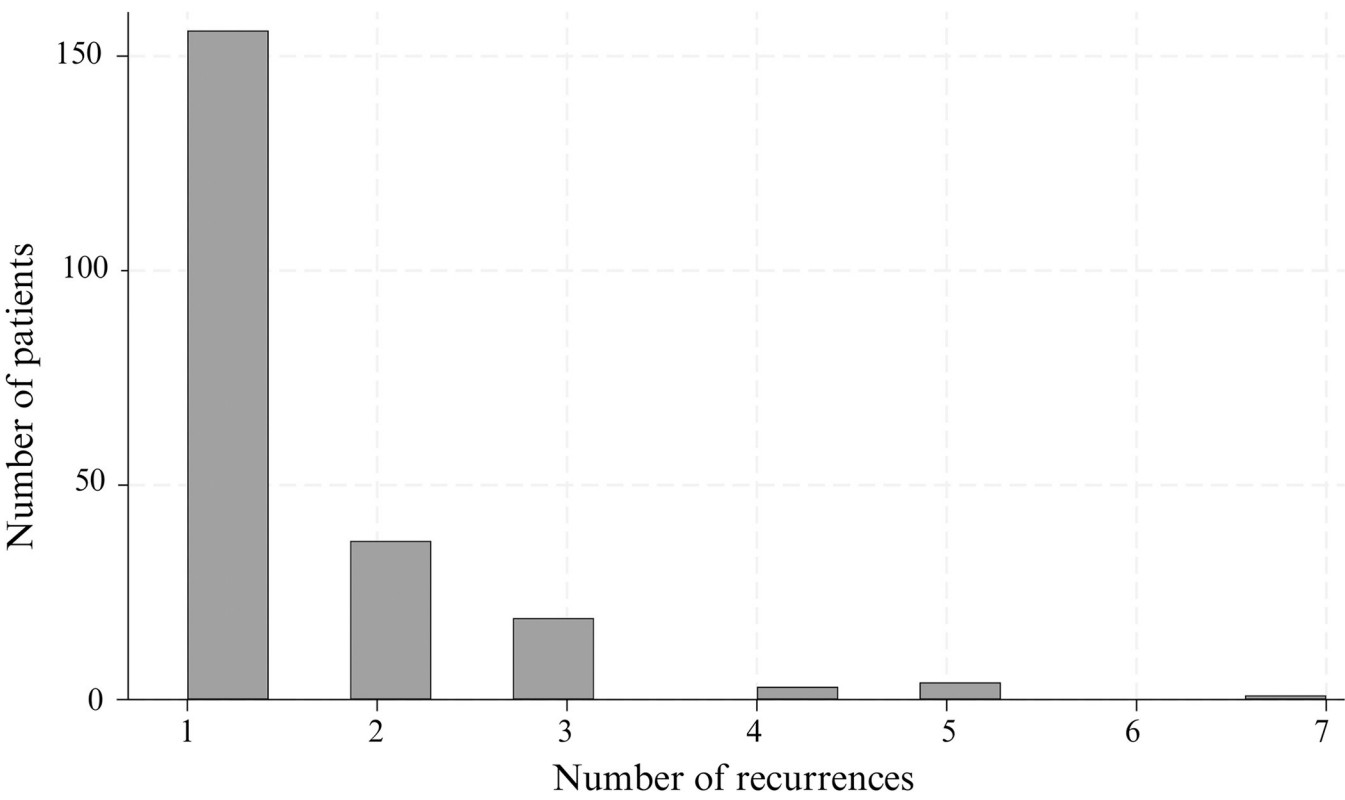

**Fig 7. Histogram of the distribution of the number of recurrent *P. vivax* infections during the observation period (January 2021 –March 2023).**

Females represented 17.5% (234/1334) of vivax patients; 70.5% (165/234) of females never received primaquine compared to 45.1% (496/1100) of males who never received primaquine (p<0.001). Females did not receive treatment either because they were not tested (131, 79.4%), were tested but then not eligible for treatment (30, 18.2%), or did not receive primaquine despite being eligible (4, 2.4%).

A district-level malaria partner acknowledged the contribution of patients who are under 20kg and female to low treatment coverage, challenging the overall elimination efforts:

*"[. . .] they don't have enough G6PD and one more thing, the pregnant women, and the children or patients who are below 20kg, they can't have the complete treatment so they can transmit the disease to others."*–cmnp1

The incidence rate of vivax malaria among patients who received radical cure for all their episodes was nine episodes per 100 person-years of observation, while patients who did not receive any primaquine had an incidence rate of 20 episodes per 100 person-years. The impact of radical cure on the reduction of vivax recurrences, enabled by the implementation of G6PD testing, was apparent to malaria officials, healthcare providers, and patients alike (po2, hc6, pp35). Healthcare providers noticed that the same patients were not presenting to the health center repeatedly, or if presenting multiple times, that the time between the recurrences was longer than previously. Malaria officials directly linked the reduction of vivax presentations to the implementation of G6PD testing (po2), while patients who had not gotten re-infected noticed that after taking complete treatment, they were less likely to get sick again, but they did not link this directly to the availability of testing (pp32).

**Table 2. Characteristics of individuals diagnosed with *P. vivax* stratified by treatment with primaquine.**

| Characteristic* | | Patients Always Primaquine | | Patients Never Primaquine | |
|---|---|---|---|---|---|
| | | Number (n = 595) | Percentage (%) | Number (n = 661) | Percentage (%) |
| Health Center | Chheu Tom | 95 | 16.0 | 45 | 6.8 |
| | Phnom Kravanh | 88 | 14.8 | 39 | 5.9 |
| | Pramoay | 87 | 14.6 | 35 | 5.3 |
| | Prongil | 19 | 3.2 | 10 | 1.5 |
| | Svay Sor | 16 | 2.7 | 2 | 0.3 |
| | Koh Preah | 45 | 7.6 | 55 | 8.3 |
| | Siem Pang | 169 | 28.4 | 437 | 66.1 |
| | Srae Kor | 42 | 7.1 | 32 | 4.8 |
| | Steung Traeng | 34 | 5.7 | 6 | 0.9 |
| Initial Point of Care | CHW | 319 | 53.6 | 539 | 81.5 |
| | Health Center | 240 | 40.3 | 122 | 18.5 |
| | Unknown | 36 | 6.1 | 0 | 0.0 |
| Age | Median | 25 (16–28, 6–76) | - | 22 (12–29, 1–67) | - |
| | Unknown | 1 | 0.2 | 2 | 0.3 |
| Weight | Under 20 kgs | 0 | 0.0 | 106 | 16.0 |
| | Over 20 kgs | 590 | 99.2 | 547 | 82.8 |
| | Unknown | 5 | 0.8 | 8 | 1.2 |
| Sex | Male | 532 | 89.4 | 496 | 75.0 |
| | Female | 63 | 10.6 | 165 | 25.0 |
| G6PD Tested | | Number (n = 476) | Percentage | Number (n = 146) | Percentage (%) |
| G6PD Status | Normal | 428 | 89.9 | 19 | 13.0 |
| | Intermediate | 2 | 0.4 | 15 | 10.3 |
| | Deficient | 43 | 9.0 | 112 | 76.7 |
| | Unknown | 3 | 0.6 | 0 | 0.0 |

*P. vivax patients who received primaquine for some of their episodes but not for all (n = 78) are not included in the table. Per definition those patients had >1 episode, while patients categorized as always or never having received primaquine had ≥ 1 episode(s).

"*If we don't test the G6PD, it is very difficult to treat the Pv patients, but now since we have this machine to test G6PD and we provide the complete treatment, we can **see that it is decreasing the cases a lot, and it will lead to malaria elimination in Pursat as well as the whole Cambodia**.*"–po2

"*Before when I got malaria, there could be a recurrence, because that time I took quinine, when I work hard and could not eat enough it can recur, **but when I took this medicine, it did not come back. [. . .] Yes, it did not come back, I am completely cured. I am thankful.**"*–pp32

## Discussion

Cambodia's national malaria program has implemented G6PD testing using the Biosensor as part of vivax case management up to the health center level. Though our study focused on the Biosensor, findings relating to the test's implementation may be applicable to and relevant for other PoC G6PD tests that might be available and introduced later in Cambodia. Our study highlights that despite a relatively complex testing procedure, the test itself has been well

accepted by malaria officials, healthcare providers, and patients—all of whom reported a positive impact of testing or testing-enabled treatment on the recurrence of vivax malaria. Healthcare providers at the health centers were able to adopt the Biosensor and provide treatment based on its results. However, an important challenge was the low referral rate (49.2%) of patients with vivax malaria from the community to the health center where G6PD testing and primaquine were provided; hence, a large proportion (62.9%) of eligible patients did not receive radical curative treatment. A lower percentage of females received radical cure compared to males, and children under 20 kilograms but over six months old were not tested and treated as per current policy. Low referral rates were likely a result of difficult road conditions and long distances to the health center, compounded by the opportunity cost of seeking further care. Patients' perceptions and prior experience with vivax malaria and the required treatment also contributed to low referral rates.

The analytical framework generated for this study included nine components—acceptability, adoption, appropriateness, feasibility, fidelity, implementation cost, sustainability, intervention expectations, and coherence operating at macro, meso, and macro levels. Collectively, these were used to assess and understand the rollout and implementation of a national intervention.

The strong adoption of the Biosensor at health centers was enabled by overcoming numerous barriers to the adoption of PoC tests that have been documented previously [25,44]. Economic, macro-level barriers [25,44], such as the cost of PoC tests, were addressed and mitigated by procurement through the Global Fund. While the sustainability of funding for test supplies may be a concern [44], within the context of elimination, supplies at this scale may not be needed for an infinite period. Policy-related obstacles at a meso-level [44], including unclear policy recommendations, were mitigated by the provision of clear guidelines from CNM. Several other macro and meso-level factors, also identified by Miller *et al.* and Pai *et al.* as being important for the adoption of PoC tests, were also overcome [25,44]. These factors included temperature stability of the test (by providing refrigerators) and a consistent supply of testing materials. A component not included in the original framework were device-level factors that contributed to the device's acceptability, such as clarity in test interpreting (*i.e.*, clear numeric output) and the availability of quality assurance for users.

In addition to the Biosensor's acceptability, micro-level factors, namely what healthcare providers expected from the intervention and how healthcare providers 'made sense' (coherence) of G6PD testing, were paramount to how the test became embedded into the routine of healthcare providers. There were expectations that the test would facilitate access to primaquine and eventually malaria elimination; and healthcare providers had a clear understanding of the specific use of the test, notably to provide safe and effective treatment. Observation of positive treatment outcomes further reinforced their understanding of the device and its purpose. Without the inclusion of coherence and expectation from normalization process theory [37] and Greenhalgh and Russel's framework [36] in addition to Proctor *et al.*'s 'implementation outcomes' [35] in our framework, key factors to the Biosensor's implementation would likely have been missed in our analysis. The importance of how interventions are understood and continuously made sense of in implementation or in practice is corroborated by similar findings from a previous study of a PoC test in a high-income context [45]. Similar to our findings, a clear understanding of the use of the PoC test and its perceived advantages enabled the adoption, despite some lack of trust in the respective test results and complexities with test processes [45].

The implementation of the Biosensor in Cambodia stands in contrast to the adoption of malaria RDTs (mRDTs), which was often protracted because of the availability of embedded alternative diagnostic mechanisms, e.g., microscopy or clinical diagnosis and supply chain

issues, among other factors [26,46–48]. In the case of the Biosensor, the lack of alternative diagnostic options and the understanding that testing enables safer treatment provision likely contributed to its faster adoption.

The implementation of the Biosensor in Cambodia has also come with strong adherence to treatment guidelines based on test results, similar to recent findings from Vietnam and Brazil [49,50]. This is different from the challenges identified with mRDTs, where health workers provided malaria treatment despite a negative mRDT result [51–54]. Among other factors, patient expectations of receiving treatment impacted health workers' decisions [27,28,51,55].

Our data suggests that embedding the Biosensor and radical cure treatment into routine practices can be understood as an equilibrium. Adoption and fidelity to guidelines are ensured by balancing the need to expand access to radical cure, with malaria elimination as the end goal, and the risk of severe drug-induced hemolysis when primaquine is prescribed without G6PD testing. The institutionalized, and sometimes individual, fear of drug-induced hemolysis, trust in the G6PD device and its results, a clear rationale for G6PD testing reinforced by observing repeated recurrences and the impact of radical cure, and a sense of duty and responsibility contributed to reaching this equilibrium.

In accordance with guidelines, healthcare providers did not administer primaquine without prior G6PD testing. The G6PD test, therefore, enables access to primaquine treatment. Providers trusted the Biosensor to improve patient safety, thus the fear of hemolysis after testing did not outweigh the anticipated positive effects of treatment with primaquine. Importantly, and a key difference to mRDT implementation [51–54], all patients were treated with schizontocidal treatment and thus did not leave health centers empty-handed regardless of the G6PD test result. Health center staff were, therefore, not incentivized to provide radical cure treatment without a G6PD normal test result to address patients' expectations for treatment provision [28]. Despite health center staff adopting the Biosensor at health center in over 90% of cases, access to G6PD testing and radical cure was still limited by significant challenges with referral from the community where patients were diagnosed with vivax malaria and provided schizontocidal treatment; less than half of eligible patients were successfully referred putting into question the overall feasibility and appropriateness of G6PD testing at the health center. Our findings are consistent with an assessment of the pilot of the Biosensor and the first phase of its implementation before the national rollout (Nov 2019-Dec 2020), where only 27% of patients were successfully referred, but 99% who reached the health center were tested [23]. The financial incentive scheme implemented in September 2022 by CNM was designed to improve referral; however, it has done so with modest effect. The low referral rates are not caused by a non-acceptance of G6PD testing itself. Instead, they are due to systemic, macro-level limitations.

Hence, improvements in infrastructure and income security are required to increase referral rates to health centers. In the absence of these improvements, a primary solution to the low referral rate would be to facilitate G6PD testing and radical cure at the community level. CHWs play a significant role in the success of Cambodia's malaria program and are essential in the elimination campaign [56–58]. CHWs continue to show interest in expanding their roles, as their responsibilities have decreased over time with the decline in malaria cases [57,59]. The use of the Biosensor by CHWs has previously been proposed [60], and an initial feasibility study was conducted in Phnom Kravanh district in 2021 and 2022 [59,61]. Twenty-eight CHWs were trained to use the Biosensor and supervised monthly by trained laboratory technicians. CHWs were confident and amenable to using the test despite some difficulties in the process, and accuracy of test results was acceptable [59,61].

Our findings of perceptions of disease and treatment suggest that a supportive strategy to incrementally increase referral and improve uptake of testing and treatment in the community

and at health centers could be a combination of more effective health communication and community engagement [62–66]. Health education can impact disease and treatment perception, while community engagement enhances ownership—both leading to increased uptake of interventions [62–66]. Such strategies, however, should consider patients' unique situations and income status and are unlikely to fully overcome structural barriers.

The content of the messaging should reflect and address patients' pre-existing understandings of radical cure and utilize perceived benefits. Our findings show that in the case of Cambodia, these benefits include aiding national goals of malaria elimination and the personal economic gains from reducing recurrences. However, messaging regarding the benefits of radical cure must be balanced with nuanced communication about the risk of reinfection and the need for continued infection prevention. Otherwise, the portrayal of primaquine as a cure or 'magic bullet' for vivax malaria could erode trust in both the treatment and healthcare providers.

The availability of PoC G6PD diagnostics enables the consideration of higher, more effective primaquine doses [67] as well as single-dose tafenoquine [68]. While the use of tafenoquine is not currently an option in Cambodia because of the drug's restriction for use with chloroquine only, ensuring adequate access to G6PD testing will be crucial for the broad roll-out of these novel treatment options.

Our study has several limitations. Firstly, complete geographic overlap between sites where qualitative and quantitative data was collected was not possible; therefore, some factors affecting implementation in some sites may have been missed or not applicable to other sites; however, we collected a large dataset including voices from a diverse and layered range of stakeholders. Furthermore, vivax patients who participated in the study all presented at a health center; therefore, perspectives on difficulties in referral to the health center were less patient-informed. Secondly, given the volume of KIIs and FGDs, multiple translators conducted translation and transcription, which could have led to different understandings of meaning and hence impacted the resulting transcripts. This was mitigated through an iterative review process and through discussions between the study team and the translators. Thirdly, there were challenges in the data collection process for the quantitative data due to data storage practice; therefore, it is possible that some vivax episodes, especially at the CHW level, are missing from our analysis. However, our data were compared with aggregated data on vivax malaria case management available at the central level, and there was a high level of agreement. Fourthly, we were not able to directly compare the impact of radical cure on recurrences between patients who always received radical cure and those who never did since the characteristics and composition of both cohorts differed. Lastly, given that our quantitative data only comes from nine sites, our findings may not reflect national trends.

## Conclusion

Overall, our results highlight the importance of including stakeholders' expectations and how they make sense of technologies in practice in the evaluation of novel interventions. Our data highlights that improved strategies are needed to ensure access to radical cure of vivax malaria, particularly for patients diagnosed at the community level. While improvements in infrastructure and income security are longer-term solutions to overcome identified barriers, more effective health communication and community engagement may encourage patients to travel to the health center to receive radical cure. However, there is an underlying need of mechanisms that bring treatment closer to patients reducing the need for referral, including equipping CHWs with G6PD tests to enable the provision of curative treatment of vivax malaria in the community.

## Supporting information

**S1 Text. Appendix A in S1 Text. Examples of KII and FGD discussion guides.** Discussion guides developed for health facility staff respondents and routine patient respondents. **Table A in S1 Text. Variables collected from health facility and community health worker forms. Table B in S1 Text. Percent completeness of routine data gathered for analysis as compared to aggregated routine surveillance data. Table C in S1 Text. List of Key Informant Interviews, Focus Group Discussions, and Observations conducted. Table D in S1 Text. Demographic characteristics of KII and FGD participants. Table E in S1 Text.** *Plasmodium vivax* **episodes for which routine and non-routine care was received from nine selected health center catchment areas in Cambodia between January 2021 and March 2023 disaggregated by year. Table F in S1 Text. Initial point of care and referral rates from the community to health centers for G6PD testing and PQ treatment disaggregated by health center. Table G in S1 Text. Identified Advantages, Challenges, and Recommendations for the STANDARD G6PD Test (Biosensor). Table H in S1 Text. G6PD status of** *Plasmodium vivax* **patients tested for G6PD deficiency at nine health centers in Cambodia. Fig A in S1 Text. Median months between recurrences of patients presenting to health centers between January 2021 and March 2023.**
(PDF)

## Acknowledgments

We are thankful to all study participants who shared their experiences and made this research possible. We also recognize the study team members who facilitated the logistics and implementation of the study. Essential to this study were also the primary translators who transcribed and translated the KIIs and FGDs from Khmer to English, including Phorn Nayelin. Finally, we acknowledge the support provided by CNM in facilitating the study and CNM's implementation partners, University Research Co (URC), Catholic Relief Service (CRS), and Clinton Health Access Initiative (CHAI), who supported data collection efforts.

## Author Contributions

**Conceptualization:** Sarah A. Cassidy-Seyoum, Dysoley Lek, Nora Engel, Kamala Thriemer.

**Data curation:** Sarah A. Cassidy-Seyoum, Keoratha Chheng.

**Formal analysis:** Sarah A. Cassidy-Seyoum, Keoratha Chheng, Benedikt Ley, Nora Engel, Kamala Thriemer.

**Funding acquisition:** Kamala Thriemer.

**Investigation:** Sarah A. Cassidy-Seyoum, Keoratha Chheng, Phal Chanpheakdey, Bipin Adhikari.

**Methodology:** Sarah A. Cassidy-Seyoum, Ric N. Price, Nora Engel, Kamala Thriemer.

**Project administration:** Sarah A. Cassidy-Seyoum, Keoratha Chheng, Rupam Tripura, Bipin Adhikari, Kamala Thriemer.

**Resources:** Lorenz von Seidlein, Rupam Tripura, Bipin Adhikari, Dysoley Lek, Kamala Thriemer.

**Supervision:** Agnes Meershoek, Michelle S. Hsiang, Bipin Adhikari, Benedikt Ley, Ric N. Price, Nora Engel, Kamala Thriemer.

**Validation:** Benedikt Ley, Nora Engel, Kamala Thriemer.

**Visualization:** Sarah A. Cassidy-Seyoum, Benedikt Ley, Ric N. Price, Nora Engel, Kamala Thriemer.

**Writing – original draft:** Sarah A. Cassidy-Seyoum.

**Writing – review & editing:** Sarah A. Cassidy-Seyoum, Keoratha Chheng, Phal Chanpheak-dey, Agnes Meershoek, Michelle S. Hsiang, Lorenz von Seidlein, Rupam Tripura, Bipin Adhikari, Benedikt Ley, Ric N. Price, Dysoley Lek, Nora Engel, Kamala Thriemer.

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
