## [Decision Letter · Decision Letter 0]

11 Jun 2024

PGPH-D-24-00551

Implementation of Glucose-6-Phosphate Dehydrogenase (G6PD) testing for *Plasmodium vivax* case management, a mixed method study from Cambodia

Dear Dr. Cassidy-Seyoum,

Thank you for submitting your manuscript to PLOS Global Public Health. After careful consideration, we feel that it has merit but does not fully meet PLOS Global Public Health’s publication criteria as it currently stands. Therefore, we invite you to submit a revised version of the manuscript that addresses the points raised during the review process.

We look forward to receiving your revised manuscript.

Kind regards,

Abhinav Sinha, M.D.

Academic Editor

Journal Requirements:

2. Please include a complete copy of PLOS’ questionnaire on inclusivity in global research in your revised manuscript. Our policy for research in this area aims to improve transparency in the reporting of research performed outside of researchers’ own country or community. The policy applies to researchers who have travelled to a different country to conduct research, research with Indigenous populations or their lands, and research on cultural artefacts. The questionnaire can also be requested at the journal’s discretion for any other submissions, even if these conditions are not met. Please find more information on the policy and a link to download a blank copy of the questionnaire here: https://journals.plos.org/globalpublichealth/s/best-practices-in-research-reporting. Please upload a completed version of your questionnaire as Supporting Information when you resubmit your manuscript.

3. We do not publish any copyright or trademark symbols that usually accompany proprietary names, eg  ©, ®, ™  (e.g. next to drug or reagent names). Please remove all instances of trademark/copyright symbols throughout the text, including ™ on page 3.

Additional Editor Comments (if provided):

Reviewers' comments:

Reviewer's Responses to Questions

**Comments to the Author**

1. Does this manuscript meet PLOS Global Public Health’s publication criteria? Is the manuscript technically sound, and do the data support the conclusions? The manuscript must describe methodologically and ethically rigorous research with conclusions that are appropriately drawn based on the data presented.

Reviewer #1: Yes

Reviewer #2: Yes

2. Has the statistical analysis been performed appropriately and rigorously?

Reviewer #1: Yes

Reviewer #2: Yes

3. Have the authors made all data underlying the findings in their manuscript fully available (please refer to the Data Availability Statement at the start of the manuscript PDF file)?

Reviewer #1: Yes

Reviewer #2: Yes

4. Is the manuscript presented in an intelligible fashion and written in standard English?

Reviewer #1: Yes

Reviewer #2: Yes

5. Review Comments to the Author

Reviewer #1: The manuscript on implementation of G6PD testing in Cambodia uses mixed methods approach to address the use of G6PD tests. Half of P. vivax diagnosis in the field are able to reach health centers to obtain testing for G6PD. Cambodia has 4,000 cases of malaria with about 90% P. vivax.

Please note in introduction or discussion that those with only 10% (deficient) of normal levels are at most risk of hemolysis requiring change in dosing of primaquine to one per week for 8 weeks and no dosing of tafenoquine.While this in figure 3, it needs to be stated.

Also explain how 437 P. vivax patients make it to the clinic yet 790 or 353 more are tested in abstract. Abstract should note that most of these had both diagnosis and G6PD testing in Health center. What is the denominator for the 353 G6PD tests that did not originate from field. The approximate numbers come out later in results abut need clarification in the abstract. Where is the 353 in the flow chart of numbers?

The perceptions are important of the utility or cost benefit for taking the 14 days of meds.

The low use in females and those under 20kg is important enough to move to abstract.

The authors should comment on a possible monthly plan to carry G6PD tests to villages to test and treat with primaquine those unable to make the journey to health center. I know there are refrigeration issues but these might be overcome

The timing of primaquine is not important after treatment with the blood stage treatment also providing post treatment prophylaxis from new blood stages for a few weeks.

The results section is full of good data.

The first paragraph of the discussion will benefit from a 500 word consise summary of important results from both quantitative and qualitative results.

Reviewer #2: Sarah Cassidy-Seyoum and colleagues present a mixed-methods study on barriers to the implementation of G6PD testing for vivax malaria treatment in Cambodia.

First, a disclaimer: I am not familiar with methods for qualitative research and suggest that a specialist in this area should also evaluate the manuscript.

However, as a malariologist, I can say that the paper is wery well written and addresses a question of major public health interest at times of tafenoquine introduction in vivax malaria treatment. Over more than five decades, primaquine has been deployed in Plasmodium vivax-endemic settings with no previous G6PD testing, but in Cambodia and other Southeast Asia the relatively high prevalence of severe G6PD deficiency variants in human populations is a major barrier to primaquine use -- and surely to tafenoquine use in the near future.

There are several findings in this study that are of interest for a broad audience of malariologists and public health specialists. For exemple, a community health worker (CHW) says: "(...) if we ask [patients] to come back to the village for the treatment [which is only available after testing], they will not come because they can’t leave their personal belongings in the forest, [...], they will not come because they earn nothing yet.” As a consequence, only 49.2% of eligible patients seen by CWHs reached health centers for G6PD testing. It remained unclear to me what happened to those patients who remained G6PD-untested? Were they treated with chloroquine alone for 3 days (either followed or not by weekly chloroquine) or left untreated? This is a key point, as the policy of requiring previous G6PD testing before providing radical vivax malaria treatment might actually prevent some patients from receiving any treatment!

Overall, the discussion is well balanced. I particularly like the following comment: "Portrayal of primaquine [I would add: also of tafenoquine] as a cure or ‘magic bullet’ for vivax malaria could erode trust in both the treatment and healthcare providers".

However, I miss comments about the barriers to the (future) implementation of tafenoquine in settings like Cambodia. Tafenoquine was the actual trigger of the renewed interest of diagnostic companies on developing point-of-care diagnostics for G6PD deficiency. If G6PD testing is not made widely available to all eligible patients, even those living in remote villages or deep in the forest, tafenoquine will never be succesfully implemented in Cambodia.

In addition, I am a bit annoyed by the tendency to equate "point-of-care G6PD testing" with a particular product (STANDARD G6PD test, SD Biosensor), although several other relatively simple (and quantitative) diagnostic methods have been developed and can be used, if not by CHWs in remote villages, at least by technicians with very basic laboratory skills in small towns and cities. Relying on a single product, from a single manufacturer, to imprement a countrywide policy of radical malaria cure seems too risky, given potential issues with timely procurement, distribution, and maintenance of the handheld devices and consumables. A comment on this topic might perhaps be appropriate, at the authors' discretion.

6. PLOS authors have the option to publish the peer review history of their article (what does this mean?). If published, this will include your full peer review and any attached files.

**Do you want your identity to be public for this peer review?** For information about this choice, including consent withdrawal, please see our Privacy Policy.

Reviewer #1: No

Reviewer #2: **Yes: **Marcelo Ferreira

---

## [Editor Report · Decision Letter 1]

21 Jun 2024

Implementation of Glucose-6-Phosphate Dehydrogenase (G6PD) testing for *Plasmodium vivax* case management, a mixed method study from Cambodia

PGPH-D-24-00551R1

Dear Sarah,

We are pleased to inform you that your manuscript 'Implementation of Glucose-6-Phosphate Dehydrogenase (G6PD) testing for *Plasmodium vivax* case management, a mixed method study from Cambodia' has been provisionally accepted for publication in PLOS Global Public Health.

Best regards,

Abhinav Sinha, M.D.

Academic Editor